# ACTOR-ATTENTION-CRITIC FOR MULTI-AGENT REINFORCEMENT LEARNING

## ABSTRACT

Reinforcement learning in multi-agent scenarios is important for real-world applications but presents challenges beyond those seen in single-agent settings. We present an actor-critic algorithm that trains decentralized policies in multi-agent settings, using centrally computed critics that share an attention mechanism which selects relevant information for each agent at every timestep. This attention mechanism enables more effective and scalable learning in complex multi-agent environments, when compared to recent approaches. Our approach is applicable not only to cooperative settings with shared rewards, but also individualized reward settings, including adversarial settings, and it makes no assumptions about the action spaces of the agents. As such, it is flexible enough to be applied to most multi-agent learning problems.

## 1 INTRODUCTION

Reinforcement learning has recently made exciting progress in many domains, including Atari games (Mnih et al., 2015), the ancient Chinese board game, Go (Silver et al., 2016), and complex continuous control tasks involving locomotion (Lillicrap et al., 2016; Schulman et al., 2015; 2017; Heess et al., 2017). While most reinforcement learning paradigms focus on single agents acting in a static environment (or against themselves in the case of Go), real-world agents often compete or cooperate with other agents in a dynamically shifting environment. In order to learn effectively in multi-agent environments, agents must not only learn the dynamics of their environment, but also those of the other learning agents present.

To this end, several approaches for multi-agent reinforcement learning have been developed. The simplest approach is to train each agent independently to maximize their individual reward, while treating other agents as part of the environment. However, this approach violates the basic assumption underlying reinforcement learning, that the environment should be stationary and Markovian. Any single agent's environment is dynamic and nonstationary due to other agents' changing policies. As such, standard algorithms developed for stationary Markov decision processes fail.

At the other end of the spectrum, all agents can be collectively modeled as a single-agent whose action space is the joint action space of all agents (Buşoniu et al., 2010). While allowing coordinated behaviors across agents, this approach is not scalable due to the action space size increasing exponentially with the number of agents. It also demands a high degree of communication during execution, as the central policy must collect observations from and distribute actions to the individual agents. In real-world settings, this demand can be problematic.

Recent work (Lowe et al., 2017; Foerster et al., 2018) attempts to combine the strengths of these two approaches. In particular, a critic (or a number of critics) is centrally learned with information from *all* agents. The actors, however, receive information only from their corresponding agents. Thus, during testing, executing the policies does not require the knowledge of other agents' actions. This paradigm circumvents the challenge of non-Markovian and non-stationary environments during learning. Despite those progresses, however, algorithms for multi-agent reinforcement learning are still far from being scalable (to a larger number of agents) and being generically applicable to environments and tasks that are co-operative (sharing a global reward), competitive, or mixed.

Our approach extends these prior works in several directions. The main idea is to centrally learn a critic with an attention mechanism. The intuition behind our idea is that in many real-world environ-

ments, it is beneficial for agents to know what other agents it should pay attention to. For example, a soccer defender needs to pay attention to attackers in their vicinity as well as the player with the ball, while she/he rarely needs to pay attention to the opposing team's goalie. The specific attackers that the defender is paying attention to can change at different parts of the game, depending on the formation and strategy of the opponent. A typical centralized approach to multi-agent reinforcement learning does not take these dynamics into account, instead simply considering *all* agents at *all* timepoints. Our attention mechanism is able to dynamically select which agents to attend to at each time point, improving performance in multi-agent domains with complex interactions.

The proposed approach has an input space linearly increasing with respect to the number of agents, as opposed to the quadratic increase in a previous approach Lowe et al. (2017). It also works well in co-operative, competitive, and mixed environments, exceeding the capability of some prior work that focuses only on co-operative environments Foerster et al. (2018).

We have validated our approach on two simulated environments and tasks. We plan to release the code for both the model and the environments after the reviewing period ends.

The rest of the paper is organized as follows. In section 2, we discuss related work, followed by a detailed description of our approach in section 3. We report experimental studies in section 4 and conclude in section 5.

## 2 RELATED WORK

Multi-Agent Reinforcement Learning (MARL) is a long studied problem (Buşoniu et al., 2010). Topics within MARL are diverse, ranging from learning communication between cooperative agents (Tan, 1993; Fischer et al., 2004) to algorithms for optimal play in competitive settings (Littman, 1994), though, until recently, they have been focused on simple gridworld environments with tabular learning methods.

As deep learning based approaches to reinforcement learning have grown more popular, they have, naturally, been applied to the MARL setting (Tampuu et al., 2017; Gupta et al., 2017), allowing multi-agent learning in high-dimensional/continuous state spaces; however, naive applications of Deep RL methods to MARL naturally encounter some limitations, such as nonstationarity of the environment from the perspective of individual agents (Foerster et al., 2017; Lowe et al., 2017; Foerster et al., 2018), lack of coordination/communication in cooperative settings (Sukhbaatar et al., 2016; Mordatch & Abbeel, 2018; Lowe et al., 2017; Foerster et al., 2016), credit assignment in cooperative settings with global rewards (Rashid et al., 2018; Sunehag et al., 2018; Foerster et al., 2018), and the failure to take opponent strategies into account when learning agent policies (He et al., 2016).

Most relevant to this work are recent, non-attention approaches that propose an actor-critic framework consisting of centralized training with decentralized execution (Lowe et al., 2017; Foerster et al., 2018), as well as some approaches that utilize attention in a fully centralized multi-agent setting (Choi et al., 2017; Jiang & Lu, 2018). Lowe et al. (2017) investigate the challenges of multi-agent learning in mixed reward environments (Buşoniu et al., 2010). They propose an actor-critic method that uses separate centralized critics for each agent which take in all other agents' actions and observations as input, while training policies that are conditioned only on local information. This practice reduces the non-stationarity of multi-agent environments, as considering the actions of other agents to be part of the environment makes the state transition dynamics stable from the perspective of one agent. In practice, these ideas greatly stabilize learning, due to reduced variance in the value function estimates.

Similarly Foerster et al. (2018) introduce a centralized critic for cooperative settings with shared rewards. Their method incorporates a "counterfactual baseline" for calculating the advantage function which is able to marginalize a single agent's actions while keeping others fixed. This method allows for complex multi-agent credit assignment, as the advantage function only encourages actions that directly influence an agent's rewards.

Attention models have recently emerged as a successful approach to intelligently selecting contextual information, with applications in computer vision (Ba et al., 2015; Mnih et al., 2014), natural

language processing(Vaswani et al., 2017; Bahdanau et al., 2015; Lin et al., 2017), and reinforcement learning (Oh et al., 2016).

In a similar vein, Jiang & Lu (2018) proposed an attention-based actor-critic algorithm for MARL. This work follows the alternative paradigm of centralizing policies while keeping the critics decentralized. Their focus is on learning an attention model for sharing information between the policies. As such, this approach is complementary to ours, and a combination of both approaches could yield further performance benefits in cases where centralized policies are desirable.

Our proposed approach is more flexible than the aforementioned approaches for MARL. Our algorithm is able to train policies in environments with any reward setup, different action spaces for each agent, a variance-reducing baseline that only marginalizes the relevant agent's actions, and with a set of centralized critics that dynamically attend to the relevant information for each agent at each time point. As such, our approach is more scalable to the number of agents, and is more broadly applicable to different types of environments.

## 3 OUR APPROACH

We start by introducing the necessary notation and basic building blocks for our approach. We then describe our ideas in detail.

### 3.1 NOTATION AND BACKGROUND

We consider the framework of Markov Games (Littman, 1994), which is a multi-agent extension of Markov Decision Processes. They are defined by a set of states, $S$, action sets for each of $N$ agents, $A_1, ..., A_N$, a state transition function, $T : S \times A_1 \times ... \times A_N \to P(S)$, which defines the probability distribution over possible next states, given the current state and actions for each agent, and a reward function for each agent that also depends on the global state and actions of all agents, $R_i : S \times A_1 \times ... \times A_N \to \mathbb{R}$. We will specifically be considering a partially observable variant in which an agent, $i$ receives an observation, $o_i$, which contains partial information from the global state, $s \in S$. Each agent learns a policy, $\pi_i : O_i \to P(A_i)$ which maps each agent's observation to a distribution over it's set of actions. The agents aim to learn a policy that maximizes their expected discounted returns, $J_i(\pi_i) = \mathbb{E}_{a_1 \sim \pi_1, ..., a_N \sim \pi_N, s \sim T}[\sum_{t=0}^{\infty} \gamma^t r_{it}(s_t, a_{1t}, .., a_{Nt})]$, where $\gamma \in [0, 1]$ is the discount factor that determines how much the policy favors immediate reward over long-term gain.

**Policy Gradients** Policy gradient techniques (Sutton et al., 2000; Williams, 1992) aim to estimate the gradient of an agent's expected returns with respect to the parameters of its policy. This gradient estimate takes the following form:

$$\nabla_\theta J(\pi_\theta) = \mathbb{E}_{a \sim \pi_\theta} \left[ \nabla_\theta \log(\pi_\theta(a_t|s_t)) \sum_{t'=t}^{\infty} \gamma^{t'-t} r_{t'}(s_{t'}, a_{t'}) \right] \tag{1}$$

**Actor-Critic and Soft Actor-Critic** The term $\sum_{t'=t}^{\infty} \gamma^{t'-t} r_{t'}(s_{t'}, a_{t'})$ in the policy gradient estimator leads to high variance, as these returns can vary drastically between episodes. Actor-critic methods (Konda & Tsitsiklis, 2000) aim to ameliorate this issue by using a function approximation of the expected returns, and replacing the original return term in the policy gradient estimator with this function. One specific instance of actor-critic methods learns a function to estimate expected discounted returns, given a state and action, $Q_\psi(s_t, a_t) = \mathbb{E}[\sum_{t'=t}^{\infty} \gamma^{t'-t} r_{t'}(s_{t'}, a_{t'})]$, learned through temporal-difference learning by minimizing the regression loss:

$$\mathcal{L}_Q(\psi) = \mathbb{E}_{s,a,r,s'} \left[ (Q_\psi(s, a) - y)^2 \right], \text{ where } y = r(s, a) + \gamma \mathbb{E}_{a' \sim \pi(s')} \left[ Q_{\bar{\psi}}(s', a') \right] \tag{2}$$

where $Q_{\bar{\psi}}$ is the target Q-value function.

To encourage exploration and avoid converging to non-optimal deterministic policies, recent approaches of maximum entropy reinforcement learning learn a soft value function by modifying the policy gradient to incorporate an entropy term (Haarnoja et al., 2018):

$$\nabla_\theta J(\pi_\theta) = \mathbb{E}_{a \sim \pi_\theta} \left[ \nabla_\theta \log(\pi_\theta(a|s))(\alpha \log(\pi_\theta(a|s)) - Q_\psi(s, a) + b(s)) \right] \tag{3}$$

where $b(s)$ is a state-dependent baseline (for the Q-value function). The loss function for temporal-difference learning of the value function is also revised accordingly with a new target:

$$y = r(s, a) + \gamma \mathbb{E}_{a' \sim \pi(s')} \left[ Q_{\bar{\psi}}(s', a') - \alpha \, \log(\pi_\theta(a'|s')) \right] \tag{4}$$

While an estimate of the value function $V_\phi(s)$ can be used a baseline, we provide an alternative that further reduces variance and addresses credit assignment in the multi-agent setting in section 3.2.

## 3.2 MULTIPLE-ACTOR-ATTENTION-CRITIC (MAAC)

The main idea behind our multi-agent learning approach is to learn the critic for each agent by selectively paying attention to other agents' actions. This is the same paradigm of training critics centrally (to overcome the challenge of non-stationary non-Markovian environments) and executing learned policies distributedly. Figure 1 illustrates the main components of our approach.

**Attention**    The attention mechanism functions in a manner similar to a differentiable key-value memory model (Graves et al., 2014; Oh et al., 2016). Intuitively, each agent queries the other agents for information about their observations and actions and incorporates that information into the estimate of its value function. This paradigm was chosen, in contrast to other attention-based approaches, as it doesn't make any assumptions about the temporal or spatial locality of the inputs, as opposed to approaches taken in the natural language processing and computer vision fields.

To calculate the Q-value function $Q_i^\psi(o, a)$ for the agent $i$, the critic receives the observations, $o = (o_1, ..., o_N)$, and actions, $a = (a_1, ..., a_N)$, for all agents indexed by $i \in \{1 \dots N\}$. We represent the set of all agents *except $i$* as $\backslash i$ and we index this set with $j$. $Q_i^\psi(o, a)$ is a function of agent $i$'s observation and action, as well as other agents' contributions:

$$Q_i^\psi(o, a) = f_i(g_i(o_i, a_i), x_i) \tag{5}$$

where $f_i$ is a two-layer multi-layer perceptron (MLP), while $g_i$ is a one-layer MLP embedding function. The contribution from other agents, $x_i$, is a weighted sum of each agent's value:

$$x_i = \sum_{j \neq i} \alpha_j v_j = \sum_{j \neq i} \alpha_j h(V g_j(o_j, a_j))$$

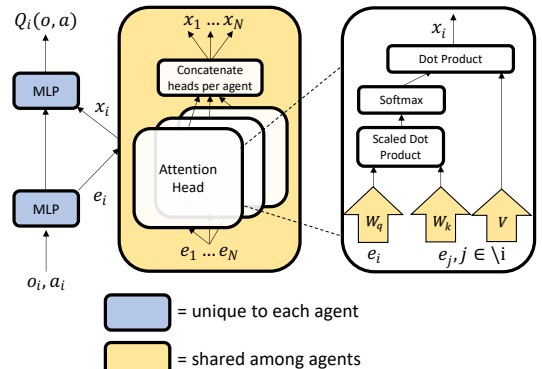

Figure 1: Calculating $Q_i^\psi(o, a)$ with attention for agent $i$.

where the value, $v_j$ is a function of agent $j$'s embedding, encoded with an embedding function and then linearly transformed by a shared matrix $V$. $h$ is an element-wise nonlinearity (we have used leaky ReLU).

The attention weight $\alpha_j$ compares the embedding $e_j$ with $e_i = g_i(o_i, a_i)$, using a bilinear mapping (ie, the query-key system) and passes the similarity value between these two embeddings into a softmax

$$\alpha_j \propto \exp(e_j^\mathsf{T} W_k^\mathsf{T} W_q e_i) \tag{6}$$

where $W_q$ transforms $e_i$ into a "query" and $W_k$ transforms $e_j$ into a "key". The matching is then scaled by the dimensionality of these two matrices to prevent vanishing gradients (Vaswani et al., 2017).

In our experiments, we have used multiple attention heads (Vaswani et al., 2017). In this case, each head, using a separate set of parameters $(W_k, W_q, V)$, gives rise to an aggregated contribution from all other agents to the agent $i$ and we simply concatenate the contributions from all heads as a single vector. Crucially, each head can focus on a different weighted mixture of agents.

Note that the weights for extracting selectors, keys, and values are shared across all agents, which encourages a common embedding space. The sharing of critic parameters between agents is possible, even in adversarial settings, because multi-agent value-function approximation is, essentially,

a multi-task regression problem. This method can easily be extended to include additional information, beyond local observations and actions, at training time, including the global state if it is available, simply by adding additional encoders, $e$. (We do not consider this case in our experiments, however, as our approach is effective in combining local observations to predict expected returns in environments where the global state may not be available).

**Learning with Attentive Critics** All critics are updated together to minimize a joint regression loss function, due to the parameter sharing:

$$
\mathcal{L}_Q(\psi) = \sum_{i=1}^{N} \mathbb{E}_{(o,a,r,o') \sim D} \left[ (Q_i^\psi(o,a) - y_i)^2 \right], \text{ where}
$$

$$
y_i = r_i + \gamma \mathbb{E}_{a' \sim \pi_{\bar{\theta}}(o')} \left[ Q_i^{\bar{\psi}}(o', a') - \alpha \, \log(\pi_{\bar{\theta}_i}(a'_i|o'_i)) \right]
$$

(7)

where $\bar{\psi}$ and $\bar{\theta}$ are the parameters of the target critics and target policies respectively. Note that $Q_i^\psi$, the action-value estimate for agent $i$, receives observations and actions for all agents. $\alpha$ is the temperature parameter determining the balance between maximizing entropy and rewards. The individual policies are updated with the following gradient:

$$
\nabla_{\theta_i} J(\pi_\theta) = \mathbb{E}_{a \sim \pi_\theta} \left[ \nabla_{\theta_i} \log(\pi_{\theta_i}(a_i|o_i))(\alpha \, \log(\pi_{\theta_i}(a_i|o_i)) - Q_i^\psi(o,a) + b(o, a_{\setminus i})) \right]
$$

(8)

where $b(o, a_{\setminus i})$ is the multi-agent baseline used to calculate the advantage function decribed in the following section. Note that we are sampling all actions, $a$, from all agents' current policies in order to calculate the gradient estimate for agent $i$, unlike in the MADDPG algorithm Lowe et al. (2017), where the other agents' actions are sampled from the replay buffer, potentially causing overgeneralization where agents fail to coordinate based on their current policies Wei et al. (2018). Full training details and hyperparameters can be found in the appendix 6.1.

**Multi-Agent Advantage Function** As shown in Foerster et al. (2018), an advantage function using a baseline that only marginalizes out the actions of the given agent from $Q_i^\psi(o,a)$, can help solve the multi-agent credit assignment problem. In other words, by comparing the value of a specific action to the value of the average action for the agent, with all other agents fixed, we can learn whether said action will cause an increase in expected return or whether any increase in reward is attributed to the actions of other agents. The form of this advantage function is shown below:

$$
A_i(o,a) = Q_i^\psi(o,a) - b(o, a_{\setminus i})), \text{ where}
$$

$$
b(o, a_{\setminus i})) = \mathbb{E}_{a_i \sim \pi_i(o_i)} \left[ Q_i^\psi(o, (a_i, a_{\setminus i})) \right]
$$

(9)

Using our attention mechanism, we can implement a more general and flexible form of a multi-agent baseline that, unlike the advantage function proposed in Foerster et al. (2018), doesn't assume the same action space for each agent, doesn't require a global reward, and attends dynamically to other agents, as in our Q-function. This is made simple by the natural decomposition of an agents encoding, $e_i$, and the weighted sum of encodings of other agents, $x_i$, in our attention model.

Concretely, in the case of discrete policies, we can calculate our baseline in a single forward pass by outputting the expected return $Q_i(o, (a_i, a_{\setminus i}))$ for every possible action, $a_i \in A_i$, that agent $i$ can take. We can then calculate the expectation exactly:

$$
\mathbb{E}_{a_i \sim \pi_i(o_i)} \left[ Q_i^\psi(o, (a_i, a_{\setminus i})) \right] = \sum_{a'_i \in A_i} \pi(a'_i|o_i) Q_i(o, (a'_i, a_{\setminus i}))
$$

In order to do so, we must remove $a_i$ from the input of $Q_i$, and output a value for every action. We add an observation-encoder, $e_i = g_i^o(o_i)$, for each agent, using these encodings in place of the $e_i = g_i(o_i, a_i)$ described above, and modify $f_i$ such that it outputs a value for each possible action, rather than the single input action. In the case of continuous policies, we do not need to add any parameters, as we can simply estimate the expectation in Equation 9 by sampling actions from our policy and averaging their Q-values, though, this comes at the cost of multiple expensive passes through the network.

## 4 EXPERIMENTS

### 4.1 SETUP

We construct two environments that test various capabilities of our approach (MAAC) and baselines. We investigate in two main directions. First, we study the scalability of different methods as the number of agents grows. We hypothesize that the current approach of concatenating all agents' observations (often used as a global state to be shared among agents) and actions in order to centralize critics does not scale well. To this end, we implement a cooperative environment, Cooperative Treasure Collection, with shared rewards where we can vary the total number of agents. The experimental results in sec 4.3 validate our claim.

Secondly, we want to evaluate each method's ability to attend to information relevant to rewards. Moreover, the relevance (to rewards) can dynamically change during an episode. This is analogous to real-life tasks such as the soccer example presented earlier. To this end, we implement a Rover-Tower task environment where randomly paired agents communicate information and coordinate.

The two environments are implemented in the multi-agent particle environment framework[1] introduced by Mordatch & Abbeel (2018), and extended by Lowe et al. (2017). We found this framework useful for creating environments involving complex interaction between agents, while keeping the control and perception problems simple, as we are primarily interested in addressing agent interaction. To further simplify the control problem, we use discrete action spaces, allowing agents to move up, down, left, right, or stay; however, the agents may not immediately move exactly in the specified direction, as the task framework incorporates a basic physics engine where agents' momentums are taken into account. Fig. 2 illustrates the two environments.

**Cooperative Treasure Collection**   The cooperative environment in Figure 2a) involves 8 total agents, 6 of which are "treasure hunters" and 2 of which are "treasure banks", which each correspond to a different color of treasure. The role of the hunters is to collect the treasure of any color, which re-spawn randomly upon being collected (with a total of 6), and then "deposit" the treasure into the correctly colored "bank". The role of each bank is to simply gather as much treasure as possible from the hunters. All agents are able to see each others' positions with respect to their own. Hunters receive a global reward for the successful collection of treasure and all agents receive a global reward for the depositing of treasure. Hunters are additionally penalized for colliding with each other. As such, the task contains a mixture of shared and individual rewards and requires different "modes of attention" which depend on the agent's state and other agents' potential for affecting its rewards.

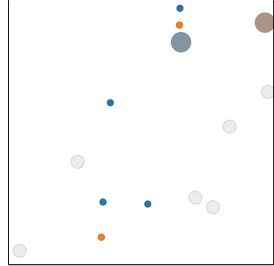

(a) Cooperative Treasure Collection. The small grey agents are "hunters" who collect the colored treasure, and deposit them with the correctly colored large "bank" agents.

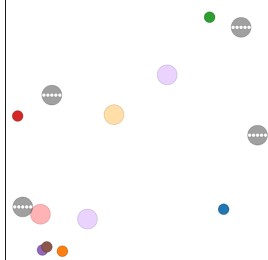

(b) Rover-Tower. Each grey "Tower" is paired with a "Rover" and a destination (color of rover corresponds to its destination). Their goal is to communicate with the "Rover" such that it moves toward the destination.

Figure 2: Our environments

**Rover-Tower**   The environment in Figure 2b involves 8 total agents, 4 of which are "rovers" and another 4 which are "towers". At each episode, rovers and towers are randomly paired. The pair is negatively rewarded by the distance of the rover to its goal. The task can be thought of as a navigation task on an alien planet with limited infrastructure and low visibility. The rovers are unable to see in their surroundings and must rely on communication from the towers, which are able to locate the rovers as well as their destinations and can send one of five discrete communication messages to their paired rover. Note that communication is highly restricted and different from centralized policy approaches Jiang & Lu (2018), which allow for free transfer of continuous information among policies. In our setup, the communication is integrated into the environment (in the tower's action

---

[1]https://github.com/openai/multiagent-particle-envs

Table 1: Comparison of various methods for multi-agent RL

|  | Base Algorithm | Attention | Centralized Critic(s) | Number of Critics | Multi-task Learning of Critics | Multi-Agent Advantage |
|---|---|---|---|---|---|---|
| MAAC (ours) | SAC | ✓ | ✓ | $N$ | ✓ | ✓ |
| MAAC (Uniform) (ours) | SAC | uniform | ✓ | $N$ | ✓ | ✓ |
| COMA[*] | Actor-Critic (On-Policy) |  | ✓ | 1 |  | ✓ |
| MADDPG[†] | DDPG |  | ✓ | $N$ |  |  |
| COMA+SAC | SAC |  | ✓ | 1 |  | ✓ |
| MADDPG+SAC | SAC |  | ✓ | $N$ |  | ✓ |
| DDPG[‡] | DDPG |  |  | $N$ | N/A | N/A |

*Centralized Critic(s)*: each agent's estimate of $Q_i$ takes the actions and observations of the other agents into account. *Number of Critics*: number of separate networks used for predicting $Q_i$ for all $N$ agents. *Multi-task Learning of Critics*: all agents' estimates of $Q_i$ share information in intermediate layers, benefiting from multi-task learning. *Multi-Agent Advantage*: cf. Sec 3.2 for details. [*](Foerster et al., 2018), [†](Lowe et al., 2017), [‡](Lillicrap et al., 2016)

space and the rover's observation space), rather than being explicitly part of the model, and is limited to a few discrete signals.

## 4.2 BASELINES

We compare to two recently proposed approaches for centralized training of decentralized policies: MADDPG (Lowe et al., 2017) and COMA (Foerster et al., 2018), as well as a single-agent RL approach, DDPG, trained separately for each agent.

In order to enable learning in discrete action spaces for both MADDPG and DDPG, where deterministic policies are not possible, we use the Gumbel-Softmax reparametrization trick (Jang et al., 2017). We will refer to these modified versions as MADDPG (Discrete) and DDPG (Discrete). For a detailed description of this reparametrization, see the appendix 6.2. We use soft actor critic to optimize. Thus, in order to have fair comparisons, we additionally implement MADDPG and COMA with Soft Actor-Critic, named as MADDPG+SAC and COMA+SAC.

We also consider an ablated version of our model as a variant of our approach. In this model, we use uniform attention by fixing the attention weight $\alpha_j$ (Eq. 6) to be $1/(N-1)$. This restriction prevents the model from focusing its attention on specific agents.

All methods are implemented such that their approximate total number of parameters (across agents) are equal to our method, and each model is trained with 6 random seeds each. Hyperparameters for each underlying algorithm are tuned based on performance and kept constant across all variants of critic architectures for that algorithm. A thorough comparison of all baselines is summarized in Table 1.

## 4.3 RESULTS AND ANALYSIS

Fig. 3 illustrates averaged rewards per episode by various methods. The proposed approach (MAAC) is competitive with other approaches being compared. In what follows, we provide detailed analysis.

**Impact of Rewards and Required Attention**  Uniform attention is competitive with our approach in the Cooperative Treasure Collection (CTC) environment, but not in Rover-Tower. On the other hand, both MADDPG (Discrete) and MADDPG+SAC perform well on Rover-Tower, though they do not on CTC. Both variants of COMA do not fare well in our environments. DDPG, arguably a weaker baseline, performs surprisingly well in CTC, but does poorly in Rover-Tower.

In CTC, the rewards are shared across agents thus an agent's critic does not need to focus on information from specific agents in order to calculate its expected rewards. Moreover, each agent's local observation provides enough information to make a decent prediction of its expected rewards. This might explain why MAAC (uniform) which attends to other agents equally, and DDPG (being very unattentive to other agents) perform well.

On the other hand, rewards in the Rover-Tower environment for a specific agent are tied to another single agent's observations. This environment exemplifies a class of scenarios where dynamic attention can be beneficial: when subgroups of agents are interacting and performing coordinated tasks with separate rewards, but the groups do not remain static. This explains why MAAC (uniform)

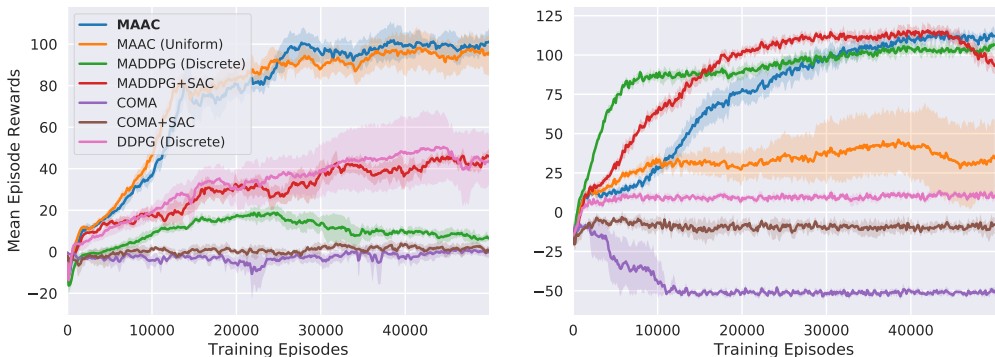

Figure 3: (Left) Average Rewards on Cooperative Treasure Collection. (Right) Average Rewards on Rover-Tower. Our model (MAAC) is competitive in both environments. Error bars are a 95% confidence interval across 6 runs.

perform poorly and DDPG completely breaks down, as knowing information from another specific agent is *crucial* in predicting expected rewards.

COMA uses a single centralized network for predicting Q-values for all agents with separate forward passes. Thus, this approach may perform best in environments with global rewards and agents with similar action spaces. However, our environments have agents with differing roles (and non-global rewards in the case of Rover-Tower). Thus both variants of COMA do not fare well.

MADDPG (and its variant) is a very strong method. However, we suspect its low performance in CTC is due to this environment's relatively large observation spaces for all agents, as the MADDPG critic concatenates observations for all agents into a single input vector for each agent's critic. Our next experiments confirm this hypothesis.

**Scalability** We compare the average rewards attained by both approaches (normalized by the range of rewards attained in the environment, as differing the number of agents changes the nature of rewards in each environment), and show that the improvement of our approach MAAC over MADDPG+SAC grows with respect to the number of agents. As suspected, MADDPG-like critics use all information non-selectively, while our approach can learn which agents to pay more attention through the attention mechanism. Thus

Table 2: MAAC improves over MADDPG+SAC

| # agents | 4 | 8 | 16 |
|---|---|---|---|
| Percentage | 20 | 49 | 53 |

our approach scales better when the number of agents increases. In future research we will continue to improve the scalability when the number of agents further increases by sharing policies among agents, and performing attention on sub-groups (of agents).

While the Rover-Tower task has a lot of agents, each agent only gets information about its paired agent – in other words, the task itself has an intrinsically smaller number of "other agents" (conditioned on each agent) than the CTC environment. As a future direction, we are creating more complicated environments where each agent needs to cope with a large group of agents where selective attention is needed. This naturally models real-life scenarios that multiple agents are organized in clusters/sub-societies (school, work, family, etc) where the agent needs to interact with a small number of agents from many groups. We anticipate that in such complicated scenarios, our approach, combined with some advantages exhibited by other approaches would do well.

## 5 CONCLUSION

We propose an algorithm for training decentralized policies in multi-agent settings. The key idea is to utilize attention in order to select relevant information for estimating critics. We analyze the performance of the proposed approach with respect to the number of agents, different configurations of rewards, and the span of relevant observational information. Empirical results are promising and we intend to extend to highly complicated and dynamic environments.

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

# 6 APPENDIX

---

**Algorithm 1** Training Procedure for Attention-Actor-Critic

---

1: Initialize $E$ parallel environments with $N$ agents, each
2: Initialize replay buffer, $D$
3: $T_{\text{update}} \leftarrow 0$
4: **for** $i_{\text{ep}} = 1 \ldots$ num episodes **do**
5:      Reset environments, and get initial $o_i^e$ for each agent, $i$
6:      **for** $t = 1 \ldots$ steps per episode **do**
7:          Select actions $a_i^e \sim \pi_i(\cdot|o_i^e)$ for each agent, $i$, in each environment, $e$
8:          Send actions to all parallel environments and get $o_i'^e, r_i^e$ for all agents in all environments
9:          Store transitions $(o_{1\ldots N}, a_{1\ldots N}, r_{1\ldots N}, o_{1\ldots N}')$ for all environments in $D$
10:          $T_{\text{update}} = T_{\text{update}} + E$
11:          **if** $T_{\text{update}} \geq$ min steps per update **then**
12:              **for** $j = 1 \ldots$ num critic updates **do**
13:                  Sample minibatch $B \leftarrow m \times (o_{1\ldots N}, a_{1\ldots N}, r_{1\ldots N}, o_{1\ldots N}') \sim D$
14:                  UPDATECRITIC($B$)
15:              **end for**
16:              **for** $j = 1 \ldots$ num policy updates **do**
17:                  Sample $m \times (o_{1\ldots N}) \sim D$
18:                  UPDATEPOLICIES($o_{1\ldots N}^B$)
19:              **end for**
20:              Update target critic and policy parameters:

$$\bar{\psi} = \tau\bar{\psi} + (1 - \tau)\psi$$

$$\bar{\theta} = \tau\bar{\theta} + (1 - \tau)\theta$$

21:              $T_{\text{update}} \leftarrow 0$
22:          **end if**
23:      **end for**
24: **end for**
25:
26: **function** UPDATECRITIC($B$)
27:      Unpack minibatch $(o_{1\ldots N}^B, a_{1\ldots N}^B, r_{1\ldots N}^B, o_{1\ldots N}'^B) \leftarrow B$
28:      Calculate $Q_i^\psi(o_{1\ldots N}^B, a_{1\ldots N}^B)$ for all $i$ in parallel
29:      Calculate $a_i'^B \sim \pi_i^{\bar{\theta}}(o_i'^B)$ using target policies
30:      Calculate $Q_i^{\bar{\psi}}(o_{1\ldots N}'^B, a_{1\ldots N}'^B)$ for all $i$ in parallel, using target critic
31:      Update critic:

$$\mathcal{L}_Q(\psi) = \sum_{i=1}^N \mathbb{E}_{(o,a,r,o')\sim D}\left[(Q_i^\psi(o, a) - y_i)^2\right], \text{ where} \tag{10}$$

$$y_i = r_i + \gamma\mathbb{E}_{a'\sim\pi_i(o')}\left[Q_i^{\bar{\psi}}(o', a') - \alpha \log(\pi_{\bar{\theta}_i}(a_i'|o_i'))\right]$$

32: **end function**
33:
34: **function** UPDATEPOLICIES($o_{1\ldots N}^B$)
35:      Calculate $a_{1\ldots N}^B \sim \pi_i^{\bar{\theta}}(o_i'^B), i \in 1 \ldots N$
36:      Calculate $Q_i^\psi(o_{1\ldots N}^B, a_{1\ldots N}^B)$ for all $i$ in parallel
37:      Update policies:

$$\nabla_{\theta_i} J(\pi_\theta) = \mathbb{E}_{a\sim\pi_\theta}\left[\nabla_{\theta_i} \log(\pi_{\theta_i}(a_i|o_i))(\alpha \log(\pi_{\theta_i}(a_i|o_i)) - Q_i^\psi(o, a) + b(o, \bar{a}))\right] \tag{11}$$

38: **end function**

---

### 6.1 TRAINING PROCEDURE

We train using Soft Actor-Critic (Haarnoja et al., 2018), an off-policy, actor-critic method for maximum entropy reinforcement learning. Our training procedure consists of performing 12 parallel rollouts, and adding a tuple of $(o_t, a_t, r_t, o_{t+1})_{1...N}$ to a replay buffer (with maximum length 1e6) for each timepoint. We reset each environment after every 100 steps (an episode). After 100 steps (across all rollouts), we perform 4 updates for the attention critic and for all policies. For each update we sample minibatches of 1024 timepoints from the replay buffer and then perform gradient descent on the Q-function loss objective (7), as well as the policy objective (8), using Adam (Kingma & Ba, 2014) as the optimizer for both with a learning rate of 0.001. These updates can be computed efficiently in parallel (across agents) using a GPU. After the updates are complete, we update the parameters $\bar{\psi}$ of our target critic $Q_{\bar{\psi}}$ to move toward our learned critic's parameters, $\psi$, as in Lillicrap et al. (2016); Haarnoja et al. (2018): $\bar{\psi} = (1 - \tau)\bar{\psi} + \tau\psi$, where $\tau$ is the update rate (set to 0.002 for attention parameters and 0.04 for all other parameters). Using a target critic has been shown to stabilize the use of experience replay for off-policy reinforcement learning with neural network function approximators (Mnih et al., 2015; Lillicrap et al., 2016). We update the parameters of the target policies, $\bar{\theta}$ in the same manner. We use a discount factor, $\gamma$, of 0.99. All networks (separate policies and contained within the centralized critics) use a hidden dimension of 128 and Leaky Rectified Linear Units as the nonlinearity. We use 0.2 as our temperature setting for Soft Actor-Critic. Additionally, we typically use 4 attention heads in our attention critics unless otherwise specified.

### 6.2 REPARAMETRIZATION OF DDPG/MADDPG FOR DISCRETE ACTION SPACES

In order to compare to DDPG and MADDPG in our environments with discrete action spaces, we must make a slight modification to the basic algorithm. This modification is first suggested by Lowe et al. (2017) in order to enable policies that output discrete communication messages. Consider the original DDPG policy gradient which takes advantage of the fact that you can easily calculate the gradient of the output of a deterministic policy with respect to its parameters.

$$\nabla_\theta J = \mathbb{E}_{s\sim\rho} \left[ \nabla_a Q(s, a)|_{a=\mu(s)} \nabla_\theta \mu(s|\theta) \right]$$

Rather than policies that deterministically output an action from within a continuous action space, we use policies that produce differentiable samples through a Gumbel-Softmax distribution (Jang et al., 2017). Using differentiable samples allows us to use the gradient of expected returns to train policies without using the log derivative trick, just as in DDPG.

$$\nabla_\theta J = \mathbb{E}_{s\sim\rho, a\sim\pi(s)} \left[ \nabla_a Q(s, a) \nabla_\theta a \right]$$

### 6.3 VISUALIZING ATTENTION

In order to understand how the use of attention evolves over the course of training, we examine the "entropy" of the attention weights for each agent for each of the four attention heads that we use in both tasks (Figures 4 and 5). The black bars indicate the maximum possible entropy (i.e. uniform attention across all agents). Lower entropy indicates that the head is focusing on specific agents, with an entropy of 0 indicating attention focusing on one agent. In Rover-Tower, we plot the attention entropy for each rover. Interestingly, each agent appears to use a different combination of the four heads, but their use is not mutually exclusive, indicating that the inclusion of separate attention heads for each agent is not necessary. This differential use of attention heads is sensible due to the nature of rewards in this environment (i.e. individualized rewards). In the case of Treasure Collection, we find that all agents use the attention heads similarly, which is unsurprising considering that rewards are shared in that environment.

In order to inspect how the attention mechanism is working on a more fine-grained level, we visualize the attention weights for one of the rovers in Rover-Tower (Figure 6), from the head that the agent appears to use the most (determined by looking at Figure 4), while changing the tower that said rover is paired to. In these plots, we ignore the weights over other rovers for simplicity since these are always near zero. We find that the rover learns to strongly attend to the tower that it is paired with, without any explicit supervision signal to do so. The model implicitly learns which

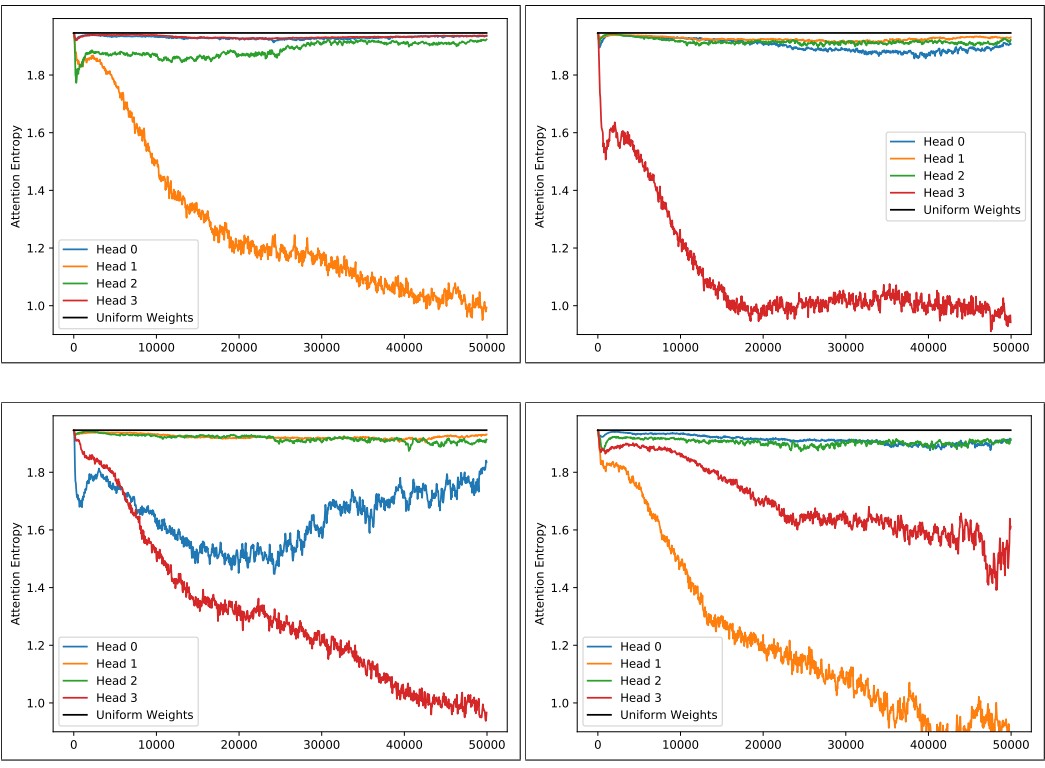

Figure 4: Attention "entropy" for each head over the course of training for the four rovers in the Rover-Tower environment

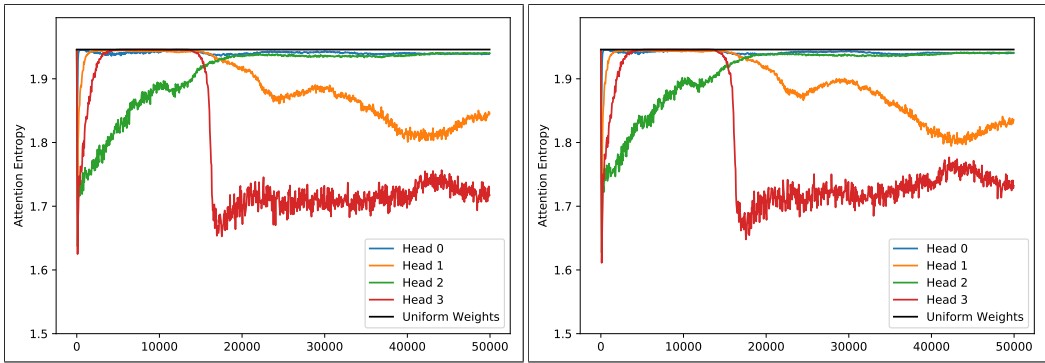

Figure 5: Attention "entropy" for each head over the course of training for two collectors in the Treasure Collection Environment

agent is most relevant to estimating the rover's expecture future returns, and said agent can change dynamically without affecting the performance of the algorithm.

## 6.4 CONTINUOUS ACTION SPACES

In order to test our model's ability to handle continuous action spaces, we add a network for each agent to learn a state-value function $V_i(o, a_{\setminus i})$, which uses the same weighted attention embedding over other agents as $Q_i(o, a)$. The loss functions to learn both networks are provided by Haarnoja et al. (2018). We test on an environment introduced in Lowe et al. (2017) called Cooperative Navigation and compare to MADDPG. Our results are presented in Table 3. This task does not require attention, as all agents are relevant to each others rewards at each time step. As such, it is unsurprising that our

Table 3: Cooperative Navigation (Continuous)

| MADDPG | MAAC |
|---|---|
| -2.47 ± 0.05 | -2.49 ± 0.11 |

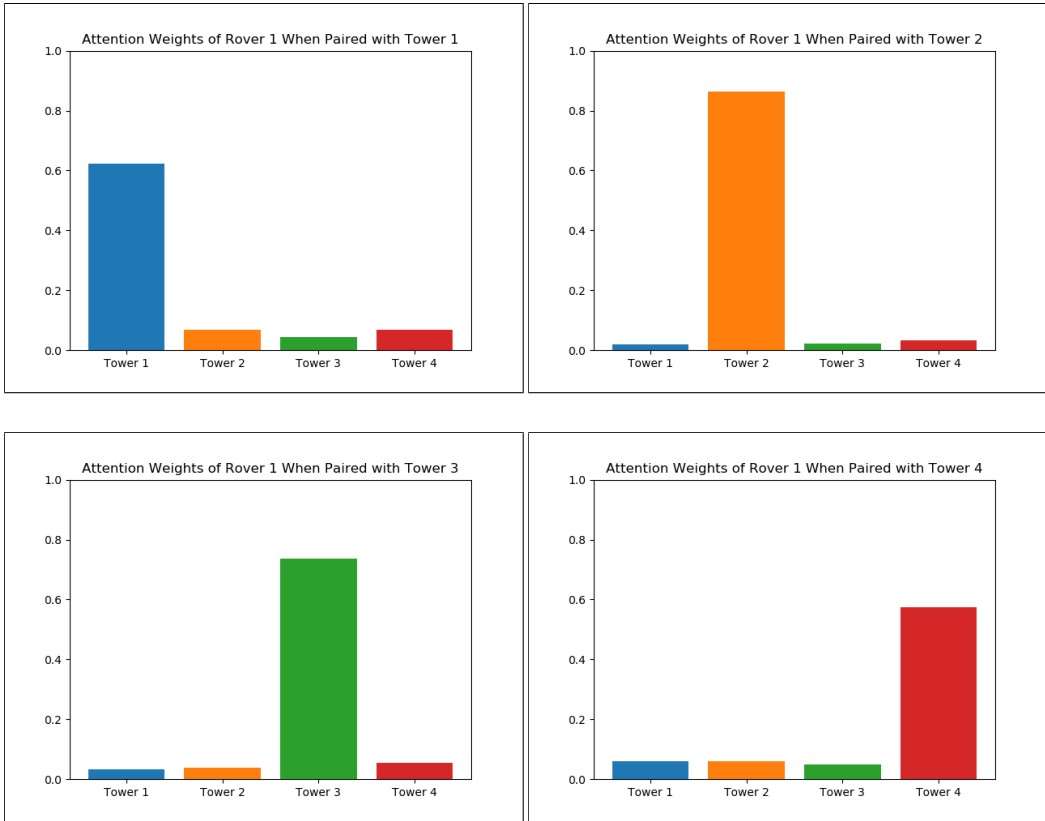

Figure 6: Attention weights when subjected to different Tower pairings for Rover 1 in Rover-Tower environment

approach matches but does not surpass the performance of MADDPG. It is notable, however, both that attention does not harm performance in simple cases and that our approach handles continuous action spaces as well.

