# OpenReview forum: "Actor-Attention-Critic for Multi-Agent Reinforcement Learning"
_ICLR.cc/2019/Conference_

### Official Review · AnonReviewer3 · 2018-10-27
**Simple method, but gives insufficient insight in model behavior and how it could generalize**

**Rating:** 4
**Confidence:** 4

**Review:**

Summary

Authors present a decentralized policy, centralized value function approach (MAAC) to multi-agent learning. They used an attention mechanism over agent policies as an input to a central value function.

Authors compare their approach with COMA (discrete actions and counterfactual (semi-centralized) baseline) and MADDPG (also uses centralized value function and continuous actions)

MAAC is evaluated on two 2d cooperative environments, Treasure Collection and Rover Tower. MAAC outperforms baselines on TC, but not on RT. Furthermore, the different baselines perform differently: there is no method that consistently performs well.

Pro
- MAAC is a simple combination of attention and a centralized value function approach.

Con
- MAAC still requires all observations and actions of all other agents as an input to the value function, which makes this approach not scalable to settings with many agents.
- The centralized nature is also semantically improbable, as the observations might be high-dimensional in nature, so exchanging these between agents becomes impractical with complex problems.
- MAAC does not consistently outperform baselines, and it is not clear how the stated explanations about the difference in performance apply to other problems.
- Authors do not visualize the attention (as is common in previous work involving attention in e.g., NLP). It is unclear how the model actually operates and uses attention during execution.

Reproducibility
- It seems straightforward to implement this method, but I encourage open-sourcing the authors' implementation.

---

> ### Author Response · Authors · 2018-11-14
> **Thank you for your comments**
>
> Thank you for your comments.  With respect to your concern over scalability, the need to input the actions and observations of all agents in the value function (i.e. centralized value function) limits scalability only during training time, and it is a necessary measure to reduce the non-stationarity of multi-agent environments, as discussed in previous work [1].
>
> We would also like to re-emphasize the fact that our final trained policies are decentralized and do not require any information exchange between agents. This trait makes our approach (and other centralized-critic/decentralized-policy approaches) useful in situations where one can train in a simulation where communication is less taxing, but deploy in the real world, where communication may be more challenging.
>
> We also compared to other methods demonstrating the better scalability of our approach, cf. Table 2.
>
> Your thinking of ‘semantically probable’ exchange of information is interesting. We note that it is possible to compress each agent’s actions/observations before they are sent to a central critic. Our setup naturally allows for this. Consider a case with high-dimensional image observations. In our approach, each agent needs to embed these observations (along with their actions) before sharing with other agents. In a situation where information exchange between agents is expensive, even during training, we can select a sufficiently small embedding space such that performance and efficiency are balanced. This notion of compressing embeddings prior to sharing across agents does not fit as naturally into the competing methods.
>
> Our experiments were especially designed to have two contrasting environments, so that we can illustrate two different aspects of multi-agent RL where we felt like the current approaches have not been able to address at the same time. Thus, it is by design that different baselines perform differently on them, as every approach has its own strengths and weaknesses.
>
> Our experiments demonstrate that our approach handles both environments well, which none of the baselines is able to do. Our experiments on Cooperative Treasure Collection demonstrate that the general structure of our attention model (even without considering dynamic attention as in our uniform attention baseline) is able to handle large observation spaces (and relatively larger numbers of agents) better than existing approaches which concatenate observations and actions from all agents together. Furthermore, our experiments on Rover-Tower demonstrate that the general model structure alone is not sufficient in all tasks, specifically those with separately coupled rewards for groups of agents, and dynamic attention becomes necessary.
>
> We have added a new section 6.3  to the supplement that includes visualizations of the attention mechanism both over the course of training and within episodes.
>
> Our code is available online and a link will be included in the paper once the anonymized review period is over.
>
> [1] Ryan Lowe, Yi Wu, Aviv Tamar, Jean Harb, OpenAI Pieter Abbeel, and Igor Mordatch. Multi-agent actor-critic for mixed cooperative-competitive environments. In Advances in Neural Information Processing Systems, pp. 6382–6393, 2017.

---

### Official Review · AnonReviewer2 · 2018-10-31
**Interesting new method, though more thorough experiments are needed**

**Rating:** 7
**Confidence:** 3

**Review:**

This paper introduces a new method for multi-agent reinforcement learning. The proposed algorithm -- which uses shared critics at training time but individual policies at test time -- makes use of a specialised attention mechanism. The benefits include better scalability (as the dependency of the inputs is linear in the number of agents, rather than quadratic), and also being more amenable to diverse reward and action structures than the previous work.

---------Quality and clarity---------
The paper is nicely written, and the ideas are developed in a clear fashion, if slightly verbose (the first 3 pages, though informative, might have been condensed a bit to make more room for the new algorithm). The problem is well-motivated and the benefits of the new algorithm are well showcased.

One negative point that does stick out is the bibliography, where papers that have been published for years (e.g. the Adam paper) are still referenced as arXiv preprints.

---------Originality and significance----------
Although attentive mechanisms have been around for a while, their use in this specific setting (learning shared critics for multi-agent RL) is, and yields desirable properties. The new algorithm opens the door for training in more complex environments, with a larger number of agents (although the number is still limited in the presented experiments).

The main issue I do see with the paper is its experimental section.
The two tasks are picked to showcase the benefits of the new approach. This does mean that the competing algorithms have to undergo significant changes (at least in the case of the DDPG-based methods), which takes away from the validity of the comparison.

Ideally, there would be at least one other task on which the other algorithms have been trained on by their respective authors. As mentioned right before Section 4, MAAC can be used on continuous action spaces at the price of increased computational cost, so this should be doable.


Overall, this is a nicely written paper which introduces an interesting new method for multi-agent RL, with promising initial results. A more thorough experimental section with slightly fairer comparisons would increase its quality significantly.

Pros
- clear paper, easy to read
- interesting application of attention mechanism to multi-agent RL
- promising initial results

Cons
- no comparison to related algorithms on tasks where they have already been evaluated externally
- the amount of workers is still quite limited in the experiments

---

> ### Author Response · Authors · 2018-11-14
> **Thank you for your comments**
>
> Thank you for your comments. We apologize for the oversight and have updated our bibliography to reference the appropriate conference publications where applicable.
>
> In the case of DDPG-based methods, we do make a slight modification in order to enable discrete action spaces; however, these modifications were first suggested by the original MADDPG paper (Lowe et al. 2017) in order to enable discrete communication action spaces. Furthermore, it seems that the released code for MADDPG by the original authors uses discrete action spaces by default (https://github.com/openai/multiagent-particle-envs/blob/master/multiagent/environment.py#L29) even for non-communication control. With that being said, we have implemented our method for continuous action spaces and find that it performs competitively with MADDPG on a cooperative task from that paper. We do not expect our approach to significantly outperform their method on their tasks, as those tasks do not necessitate the use of attention (all agents are generally relevant to each agent’s rewards at every time step).  The results can be seen in section 6.4 of the appendix in our revised draft.
>
> In our experiments, we use up to 16 agents.  We can further scale up, for example, using some ideas from existing works, including assuming homogenous agents and global rewards which allow for shared critics, etc.  Note that, even without those simplifications, our approach is still able to scale better than an approach, MADDPG+SAC, that follows a similar paradigm as ours (Table 2 on page 8).

---

### Official Review · AnonReviewer1 · 2018-11-05
**Interesting contribution to multiagent RL**

**Rating:** 6
**Confidence:** 3

**Review:**

The paper considers an actor-critic scheme for multiagent RL, where the critic is specific to each agent and has access to all other agents' embedded observations. The main idea is to use an attention mechanism in the critic that learns to selectively scale the contributions of the other agents.

The paper presents sufficient motivation and background, and the proposed algorithmic implementation seems reasonable. The proposed scheme is compared to two recent algorithms for centralized training of decentralized policies, and shows comparable or better results on two synthetic multiagent problems.

I believe that the idea and approach of the paper are interesting and contribute to the multiagent learning literature.

Regarding cons:
- The critical structural choices (such as the attention model in section 3.2) are presented without too much justification, discussion of alternatives, etc.
- The experiments show the learning results, but do not provide a peak "under the hood" to understand the way attention evolved and contributed to the results.
- The experiments show good results compared to existing algorithms, but not impressively so.

---

> ### Author Response · Authors · 2018-11-14
> **Thank you for your comments**
>
> Thank you for your comments. With regard to the structural choices of the attention model, our decision was based on a survey of attention-based methods used across various applications and their suitability for our problem setting. Our mechanism was designed such that, given a set of independent embeddings, each item in the set can be used to both extract a weighted sum of the other items as well as contribute to the weighted sums that other items extract. When applied to multi-agent value-function approximation, each item can belong to an agent and the separate weighted sums can be used to estimate each agent’s expected return. Some other choices of attention mechanisms such as RNN-based ones (widely used in NLP), while interesting, do not naturally extend to our setting as our inputs (ie embeddings from agents) do not form a natural temporal order. We have updated our draft to provide more insight into our choices.
> We have included a new section 6.3 in the appendix of our revised draft that visualizes the behavior of our attention mechanism, as well as how it evolves over the course of training.
>
> While our approach does not significantly outperform the best individual baseline in each environment, it consistently performs near the top in all environments --- other methods falter in at least one of the two settings. Our experiments on Cooperative Treasure Collection demonstrate that the general structure of our attention model (even without considering dynamic attention as in our uniform attention baseline) is able to handle large observation spaces (and relatively larger numbers of agents) better than existing approaches which concatenate observations and actions from all agents together. Furthermore, our experiments on Rover-Tower demonstrate that the general model structure alone is not sufficient in all tasks, specifically those with separately coupled rewards for groups of agents, and dynamic attention becomes necessary.

---

### Public Comment · (anonymous) · 2018-10-09
**Policy Gradient Theorem Holds for POMDP?**

In this work, the authors propose an actor-critic algorithm for multi-agent POMDP. The algorithm depends on the policy gradient theorem in the POMDP setting. In equation (1) the authors summarize the policy gradient theorem for MDP, however, this result does not hold for POMDP. The policy update step, namely, equation (8) lacks substantiation. It would be great if the authors could discuss more the validity of equation (8).

A related paper:
ACCNet: Actor-Coordinator-Critic Net for "Learning-to-Communicate" with Deep Multi-agent Reinforcement Learning. Mao et. al.

---

> ### Author Response · Authors · 2018-10-10
> **Thank you for the comment**
>
> We thank you for your comment. The approach of utilizing policy gradients in partially observed multi-agent environments has been established in the literature, provided that a centralized critic receives the global state [1] or the local observations of all agents [2]. The paper that you bring up (ACCNet) also appears to follow this paradigm.
>
> The authors of [1] provide a proof (pages 4-5) which shows that the multi-agent policy gradient (w/ a state-dependent baseline) reduces to the standard single agent policy gradient, provided that the combined observation histories of each agent combine to form the global state (Eqn. 15). Our approach makes a similar assumption, such that the combined observations of all agents (o = {o_1, …, o_N}) represents the global state, and this holds true for the environments that we test in. We apologize for the lack of clarity regarding this subject in the initial submission. We will revise our draft once the rebuttal period opens to reflect this point.
>
> [1] Jakob Foerster, Gregory Farquhar, Triantafyllos Afouras, Nantas Nardelli, and Shimon Whiteson. Counterfactual Multi-Agent policy gradients. arXiv preprint arXiv:1705.08926, May 2017a
>
> [2] Ryan Lowe, Yi Wu, Aviv Tamar, Jean Harb, OpenAI Pieter Abbeel, and Igor Mordatch. Multi-agent actor-critic for mixed cooperative-competitive environments. In Advances in Neural Information Processing Systems, pp. 6382–6393, 2017.

---

### Meta-Review · Area_Chair1 · 2018-12-16

**Confidence:** 4
**Recommendation:** Reject

**Metareview:**

The authors propose an approach for a learnt attention mechanism to be used for selecting agents in a multi agent RL setting. The attention mechanism is learnt by a central critic, and it scales linearly with the number of agents rather than quadratically. There is some novelty in the proposed method, and the authors clearly explain and motivate the approach. However the empirical evaluation feels quite limited and does not show conclusively that the method is superior to the others. Moreover, the simple empirical results don't give any evidence how the attention mechanism is working or whether it is truly the attention that is affecting the results. The reviewers were split on their recommendation and did not come to a consensus. The AC feels that the paper is not quite strong enough and encourages the authors to broaden the work with additional experiments and analysis.